# Comparative Brain Proteomic Analysis between Sham and Cerebral Ischemia Experimental Groups

**DOI:** 10.3390/ijms25147538

**Published:** 2024-07-09

**Authors:** María Candamo-Lourido, Antonio Dopico-López, Esteban López-Arias, Sonia López-Amoedo, Clara Correa-Paz, María Pilar Chantada-Vázquez, Ana Bugallo-Casal, Lucía del Pozo-Filíu, Lara Pérez-Gayol, Nuria Palomar-Alonso, Susana B. Bravo, Francisco Campos, María Pérez-Mato

**Affiliations:** 1Translational Stroke Laboratory Group (TREAT), Clinical Neurosciences Research Laboratory (LINC), Health Research Institute of Santiago de Compostela (IDIS), 15706 Santiago de Compostela, Spain; maria.candamo.lourido@sergas.es (M.C.-L.); antonio.dopico.lopez@sergas.es (A.D.-L.); esteban.lopez.arias@sergas.es (E.L.-A.); sonia.lopez.amoedo@sergas.es (S.L.-A.); clara.correa@hotmail.es (C.C.-P.); ana.isabel.bugallo.casal@sergas.es (A.B.-C.); lara.perez.gayol@sergas.es (L.P.-G.); nuria.palomar.alonso@sergas.es (N.P.-A.); 2Proteomic Unit, Research Institute of Santiago de Compostela (IDIS), Complejo Hospitalario Universitario de Santiago de Compostela (CHUS), 15706 Santiago de Compostela, Spain; mariadelpilarchantadavazquez@gmail.co (M.P.C.-V.); susana.belen.bravo.lopez@sergas.es (S.B.B.)

**Keywords:** animal models, experimental design, proteomic analysis, sham, stroke

## Abstract

Sham control groups are essential in experimental animal studies to reduce the influence of surgical intervention. The intraluminal filament procedure is one of the most common models of middle cerebral artery occlusion (MCAO) used in the study of brain ischemia. However, a sham group is usually not included in the experimental design of these studies. In this study, we aimed to evaluate the relevance of the sham group by analyzing and comparing the brain protein profiles of the sham and MCAO groups. In the sham group, 98 dysregulated proteins were detected, compared to 171 in the ischemic group. Moreover, a comparative study of protein profiles revealed the existence of a pool of 57 proteins that appeared to be dysregulated in both sham and ischemic animals. These results indicated that the surgical procedure required for the intraluminal occlusion of the middle cerebral artery (MCA) induces changes in brain protein expression that are not associated with ischemic lesions. This study highlights the importance of including sham control groups in the experimental design, to ensure that surgical intervention does not affect the therapeutic target under study.

## 1. Introduction

According to the World Health Organization, stroke is the second most common cause of death and disability among adults worldwide. With an aging population, the incidence of stroke in developing countries is expected to continue to increase, becoming the leading cause of premature death and disability in adults. Therefore, developing safe and effective treatments remains a major challenge in both experimental and clinical neuroscience [1].

Animal models are important tools for emulating stroke and can be used to investigate stroke mechanisms and develop new therapeutic agents. In recent years, several animal models, including intraluminal monofilament occlusion, transcranial occlusion, photothrombosis, thromboembolic occlusion, and endothelin parenchymal brain injection, have been developed to study ischemic mechanisms and drug testing [2,3]. However, many neuroprotective therapies that have shown beneficial effects in preclinical analyses ultimately fail in clinical trials [4]. There are numerous reasons for translational failure in preclinical stroke research conditioned by the experimental setting itself, such as the generation of a plausible hypothesis, the methodological quality of the experimental procedure, and the analysis of the results (ideally double-blinded). To improve the quality of preclinical studies, the Stroke Therapy Academic Industry Roundtable (STAIR) and Animal Research: Reporting of in vivo Experiments (ARRIVE) guidelines have been published [5,6]. Among the aforementioned recommendations, the importance of including a sham group stands out [7]. A sham group was defined as a group undergoing a simulated procedure, to ensure that they experienced the same side effects as the group undergoing the actual surgery or procedure. Sham groups have the potential advantage of reducing bias and are used in experimental designs to help researchers determine the effectiveness of a drug or treatment under investigation and whether the difference between the groups is caused by the surgical procedure itself [8].

Of all the models of cerebral ischemia, the intraluminal model of middle cerebral artery occlusion (MCAO) is the most widely used, because it allows the restoration of blood flow after the induction of focal ischemia, exhibits an ischemic penumbra, is highly reproducible, and does not require a craniotomy [9]. Despite the characteristics of the surgical procedure for the intraluminal filament model and the recommendations of the ARRIVE guidelines, many studies have not included a sham group in their experimental design. A Pubmed search for “intraluminal filament” in the last 5 years (2019–2024) obtained 57 results, while “intraluminal filament and sham group” returned 8, indicating that only 14% of preclinical studies used a sham group. In addition, in articles including the sham group, the surgical procedures have not been properly addressed or described, since the procedure consists only of the isolation of the common carotid artery (CCA) from the external carotid artery (ECA), underestimating the effect of permanent occlusion of the CCA to the brain, which is required for intraluminal occlusion of the middle cerebral artery (MCA) [2,10]. This evidence indicates that there is no clear consensus regarding the inclusion of a sham group in the experimental design. Therefore, we analyzed the brain protein profile of sham and ischemic animals to evaluate the impact of the surgical procedure on the target organ. The results were compared with those of a healthy control group without any surgical intervention. The sham control group underwent the same surgical procedure and anesthesia but without intraluminal occlusion of the MCA. Liquid chromatography–mass spectrometry (LC-MS/MS) and sequential window acquisition of all theoretical mass spectra (SWATH–MS) using a hybrid quadrupole time-of-flight mass spectrometer were used for the protein analysis.

## 2. Results

### 2.1. Magnetic Resonance Imaging (MRI) Analysis of Brain Tissue in Control, Sham, and Ischemic Animals

Arterial brain circulation and brain tissue were evaluated in the three groups using MRI. Angiographic imaging showed the intact arterial tree of the brain in control animals, including the anterior, middle, and posterior cerebral arteries (left and right regions) (Figure 1A), without any signs of ischemic lesions, as evaluated by T2 imaging at 24 h (Figure 1B). The sham group underwent the same surgical procedure as the ischemic animals but without intraluminal occlusion of the MCA. Angiographic imaging indicated that the cerebral vasculature was not altered, and no ischemic lesions were observed either at 24 h. Angiography demonstrated successful occlusion of the MCA associated with an ischemic lesion, evaluated using T2 imaging at 24 h, in the ischemic group (infarct volume 39.73 ± 15.15%, established as the percentage of ischemic damage with respect to the ipsilateral hemisphere volume, corrected for brain edema).

### 2.2. Protein Analysis of Brain Tissue and Qualitative Protein Analysis

Animals were subjected to MRI scans and sacrificed 24 h after surgery, and the brain tissue of the three experimental groups (control, sham, and ischemic group) were analyzed using proteomic analysis. We performed an initial qualitative analysis to identify the proteins expressed in each brain tissue using LC-MS/MS in the data-dependent acquisition (DDA) mode. We selected proteins with a false discovery rate (FDR) < 1% for each sample to obtain representative proteins per group [11]. Subsequently, a quantitative analysis was performed using the SWATH–MS method.

In this study, 1625 proteins, 36,131 distinct peptides, and 87,128 spectra were identified (FDR < 1% in each sample). The Venn diagram (Figure 2A) shows the distribution of these proteins and indicates those that were unique and common among the three experimental groups. From this analysis, 21 common proteins (see Appendix A) were detected in both the sham and ischemic groups. These common proteins were then subjected to functional protein association network analysis using STRING software (Version: 12.0). The results are summarized in Figure 2B according to the biological processes, cellular components, and molecular functions in which they are involved. The samples were subsequently subjected to principal component analysis (PCA), and the results indicated that principal component 1 (PC1) could explain 49% of the variance in proteomic changes separating the groups (Figure 2C).

### 2.3. Quantitative Protein Analysis Using SWATH–MS

A quantitative analysis was performed using the SWATH–MS method to compare the protein expression profiles of sham vs. control and ischemia vs. control samples. In this study, 1240 proteins, 10,341 distinct peptides, and 61,306 spectra with an FDR of <1% in each sample were identified. The most dysregulated proteins were identified according to a fold change (FC) ≥ 1.5 or ≤0.6, *p* ≤ 0.05.

In the first analysis (sham vs. control), 98 dysregulated proteins were identified, of which 48 were downregulated and 50 were upregulated in the sham group compared with the control group (Appendix A). A volcano plot (Figure 3A) shows the differences in protein expression between the sham and control groups. STRING software and a Gene Ontology (GO) pathway enrichment analysis indicated that the upregulated proteins were mostly involved in metabolism-related processes (intracellular cGMP-activated cation channel activity, energy derivation by the oxidation of organic compounds, glycogen metabolic processes, and glucagon and insulin signaling pathways). The downregulated proteins were primarily involved in the regulation of apoptosis, neuronal development, and axon regeneration (Figure 3B). It was observed that the main upregulated proteins (*p* < 0.05; FC ≥ 2), GLRX3, KCC2D, and NEUG, had a low expression, while TBB3 and AT1B2 were in higher abundance. A statistical analysis revealed that the expression levels of KCC2D, NEUG, and TBB3 were significantly higher than those in the controls (Figure 3C). The abundance of the most important downregulated proteins (*p* < 0.05; FC ≤ 0.5), PRPS1, ZNT3, NU5M, CSPG5, ESTD, MIF, GPM6B, ADT2, and FXYD7, was low, while MPCP, NFH, VDAC 1/2, MOG, ATP5H, ALBU, M2OM, GPM6B, ADT2, MYPR, GBB1, ADT1, HBB1, and HBA were highly expressed. The expression levels of GPM6B, ADT2, FX4D7, and MYPR were significantly lower than those in the control group (Figure 3D). The SWATH–MS areas of the main dysregulated proteins (sham vs. control) are specified in Appendix A.

A comparative protein analysis of the ischemic vs. control groups revealed 171 dysregulated, 94 downregulated, and 77 upregulated proteins (Appendix A). A volcano plot (Figure 4A) shows the differences in protein expression between the ischemic vs. control groups. Using STRING software and a GO pathway enrichment analysis, we determined that the upregulated proteins were implicated in biological processes and cellular components, such as response to stress, regulation of catalytic activity, regulation of cell death, and apoptotic processes. The downregulated proteins were primarily involved in glial cell differentiation and various metabolic processes (mitochondrial transmembrane transport, proton motive force-driven mitochondrial ATP synthesis, and oxidative phosphorylation (Figure 4B). The most important upregulated proteins (*p* < 0.05; FC ≥ 2) (FIBB, HSPB1, CO3, VTDB, SCG2, KCC2D, UNC5B, ARMT1, CRP, RLA2, NEUG, and FETUA) were less expressed; while ALBU, HS71B, A1AT, HEMO, TRFE, HBB1, HBA, KPYM, PCS1N, A1I3, and ENOA were in higher abundance. The expression levels of FIBB, CO3, SCG2, FETUA, and ALBU were significantly higher than those in the control (Figure 4C). The downregulated proteins, including H15, CNKR2, PSD11, H10, CLD11, H31, and FXYD7, were poorly expressed compared to QCR1, KCC2B, QCR2, GBB1, PHB, ADT2, AT1B2, SYNPO, GPM6A, ATPA, H14, M2OM, ATPB, MBP, VDAC1, ADT1, MOG, NFL, ATPO, VDAC2, CXA1, AINX, NFH, MYPR, CSN1, and NFM, which presented a high expression. The results showed that the decreased expression of the H10, CLD11, H31, FXYD7, AINX, and MYPR proteins compared to those in the control was statistically significant (Figure 4D). The SWATH–MS-normalized areas of the main dysregulated proteins (ischemia vs. control) are shown in Appendix A.

Finally, we analyzed the common dysregulated proteins identified using SWATH–MS in the sham and ischemic groups. Following a similar analytical procedure, the Venn diagram (Figure 5A) shows the distribution of proteins in each group, indicating that 57 common proteins appeared to be affected in both groups. The common proteins were subjected to functional protein association networks using STRING software. The results are summarized in Figure 5B according to the biological processes, cellular components, and molecular functions involved. The volcano plot shows the common dysregulated proteins in the sham (Figure 5C) and ischemic groups (Figure 5D) (Appendix A). The results demonstrated that of the 57 common proteins, 18 (KCC2D, KPYM, PCS1N, ARMT1, ENOA, RLA2, NEUG, STMN1, PTMS, TBCA, PCP4, CBPQ, PUR9, THIO, ATX10, TEBP, AP2S1, and WIPF3) were upregulated (*p* < 0.05; FC ≥ 1.5), and 35 (SUCA, GPM6B, MPCP, SFXN5, AT1A2, NMDZ1, NDUA9, ZNT3, UCRI, VDAC3, NDUV2, SEC13, DLG2, CSPG5, SFXN1, ATP5H, NU5M, ATP4A, PRPS1, SDHB, QCR1, QCR2, GBB1, PHB, ADT2, GPM6A, M2OM, VDAC1, ADT1, MOG, VDAC2, NFH, MYPR, NFM, and FXYD7) were downregulated (*p* < 0.05; FC ≤ 0.6) in both groups. Using STRING software and a GO pathway enrichment analysis, we determined that the commonly upregulated proteins were involved in prostaglandin biosynthetic processes, cellular senescence, ferroptosis, and tubulin complex assembly. The commonly downregulated proteins were implicated in axon regeneration, mitochondrial transmembrane transport, neurofilament bundle assembly, and the regulation of apoptotic processes (Figure 5E).

We also analyzed whether common dysregulated proteins were detected, using a qualitative proteomic analysis. For this purpose, we performed a comparison between the common dysregulated proteins in the sham and ischemic groups with respect to the common proteins in the control, sham, and ischemic groups (1417 proteins; Figure 2A) and the common proteins between the sham and ischemic groups (21 proteins; see Figure 2A) identified using SWATH–MS. Fifty-three proteins common to the sham and ischemic groups were found in the pool of proteins common to all three groups (Appendix A). Using two different techniques, we confirmed the existence of a significant number of common proteins between the sham and ischemic groups.

## 3. Discussion

The MCA is one of the most common vessels affected in clinical ischemic stroke; therefore, intraluminal occlusion of this artery is the most commonly used experimental approach in animal models to replicate clinical conditions. The intraluminal filament occlusion in the MCA animal model was initially developed by Koizumi et al. in 1986 and modified by Longa et al. in 1989 [12]. In this model, access to the MCA required the introduction of a filament into the internal carotid artery (ICA) through the ECA. Infarcts induced using this approach often cause striatal and cortical damage. Temporary insertion of a filament into the MCA, which is later removed after a predefined period of ischemia, produces transient MCA-territory ischemia, followed by the restoration of blood circulation. Alternatively, leaving the filament in the MCA can be used to reproduce permanent stroke without reperfusion.

The main advantages of this method include the reliability of reproducing the pathophysiology of stroke, the ability to avoid a craniotomy, and the potential for high-throughput drug screening. Although this is the most common model used in experimental stroke fields, the main disadvantages include tracheal edema, and paralysis of the muscles of mastication and swallowing caused by injury to the CCA, ICA and ECA. In addition, the clamping of the carotid arteries, which is necessary to introduce the filament into the vessels, causes a reduction in cerebral flow, which could contribute to brain injury beyond the ischemia caused by the occlusion of the MCA. Considering that the surgical procedure itself can interfere with the final results, it is particularly important to include a sham control group in preclinical studies [12]. Some studies support this premise, as they have observed that surgical procedures are not harmless to the tissue. Thus, the effects of sham surgical procedures and anesthesia on the expression patterns of microRNAs in rat liver were investigated in a study on liver regeneration [13]. They found that 49 microRNAs were modified by hepatectomy and 45 by a sham laparotomy, with 10 microRNAs showing similar changes after both real and sham surgeries. The impact of sham surgery was also highlighted by Cole et al. [14], who compared the effects of standard sham procedures used in research on traumatic brain injury (craniotomy by drill or manual trepanation) with those of anesthesia alone. They found that the traditional sham control induced significant pro-inflammatory, morphological, and behavioral changes, which could confound the interpretation of brain injury models.

In line with these previous findings, our qualitative proteomic analysis revealed the existence of 18 proteins exclusively present in the sham group and 21 common proteins present in both the sham and ischemic groups. A quantitative analysis showed 98 dysregulated proteins in the sham vs. control and 171 dysregulated proteins in the ischemia vs. control group. This was in accordance with Rutkai et al. [15], who reported variations in cerebrovascular function and mitochondrial bioenergetics in the sham MCAO group. However, most studies that performed proteomic analyses of tissues from the MCAO sham group found no proteome modifications [16,17]. This may be because those studies assumed that the sham group is a fully healthy control, when a non-interventional control group should be included in the analysis, as was done in this study. We also observed that the number of dysregulated proteins was higher in the ischemic group than in the sham group, which could be related to increased activation of pathways related to apoptotic processes, stress and cell death.

We later performed an analysis of common dysregulated proteins in the sham and ischemic group, and the following common proteins were found: ALBU, HBB1, KCC2D, HBA, KPYM, PCS1N, ARMT1, ENOA, RLA2, NEUG, STMN1, PTMS, TBCA, PCP4, CBPQ, PUR9, THIO, ATX10, TEBP, AP2S1, WIPF3, SUCA, GPM6B, MPCP, SFXN5, AT1A2, NMDZ1, NDUA9, ZNT3, UCRI, VDAC3, NDUV2, SEC13, DLG2, CSPG5, SFXN1, ATP5H, NU5M, ATP4A, PRPS1, SDHB, QCR1, QCR2, GBB1, PHB, ADT2, AT1B2, GPM6A, M2OM, VDAC1, ADT1, MOG, VDAC2, NFH, MYPR, NFM, and FXYD7. Functional enrichment and interaction network analyses demonstrated that these proteins are involved in processes related to cerebral ischemia, such as prostaglandin synthesis [18], ferroptosis [19], cellular senescence [20], and axonal regeneration [21]. These results were consistent with those of previous studies describing the involvement of the most commonly dysregulated proteins in the pathophysiology and prognosis of cerebral ischemia. Accordingly, ALBU may potentially aid in identifying patients with a subarachnoid hemorrhage at risk of developing delayed cerebral ischemia [22]. HBB1 and HBA are upregulated during hypoxia [23]. KCC2D and KPYM are key to mitochondrial function and the regulation of ferroptosis-related diseases such as stroke, sepsis, diabetic kidney disease, and tumors in mouse models [24]. ENOA and NEUG are promising tools for predicting ischemic stroke outcomes [25,26]. STMN1 is associated with neurons undergoing ectopic chain migration into the ischemic striatum and cerebral cortex following focal cerebral ischemia [27]. PHB1 is involved in the reduction of focal cerebral ischemic injury and the preservation of neurological function. The mechanisms of protection remain to be fully elucidated, but they may involve the maintenance of mitochondrial integrity, suppression of ROS production, and early pathophysiological events in ischemic stroke [28]. MOG is associated with naïve T cells and CD8+TEMRA cells in patients with acute ischemic stroke [29]. NFH and NFM are markers of neuroaxonal damage, and there is an association between these proteins and neurological and functional outcomes after stroke [30]. MYPR is the major myelin protein in the central nervous system and plays an important role in the formation and maintenance of the multilamellar structure of myelin. Oligodendrocytes (OLs) and white matter fibers are injured after stroke, which is one of the common causes of neurological dysfunction in adults [31,32].

Beyond the molecular mechanism in which these dysregulated proteins could be involved [18], these commonly affected proteins in both sham and ischemic conditions indicated that surgical intervention in the MCAO model induces changes in brain proteins, which could lead to an overestimation of the molecular mechanisms caused by ischemic damage.

A limitation of this study could be the impact of anesthesia exposure on protein expression, which was not evaluated [33,34]. Although we cannot exclude a possible interaction of this parameter, a similar exposure to sevoflurane was used in all groups (control, sham, and ischemic), demonstrating that the differences observed were not due to anesthesia.

In conclusion, it is widely assumed that surgical intervention to access the MCA has a minimal impact on cerebral tissues, and ischemic damage is only due to occlusion of the cerebral artery [8]. Our results provide clear evidence that the surgery required to induce MCA occlusion induces protein expression changes in the brain, supporting the need to include sham groups in the experimental designs of preclinical studies. These sham controls are not neutral and are highly recommended to determine how these interventions interfere with the primary purpose of the study.

## 4. Materials and Methods

### 4.1. Animal Care and Housing

The protocols for the rodent assays were approved by the Health Research Institute of Santiago de Compostela (IDIS) Animal Care Committees under procedure numbers 15011/2022/003 and 15011/2023/001, according to European Union (EU) guidelines (86/609/CEE, 2003/65/CE, and 2010/63/EU) and within the ARRIVE guidelines. Male Sprague Dawley rats (7–8 weeks old) weighing 250–300 g were used. Animals were housed at an environmental temperature of 23 °C with 40% relative humidity and a 12 h light–dark cycle. The rats were provided water and food ad libitum. The animals (two animals/box) were acclimated to the new environment for one week after arrival at the animal facility and prior to the experimental procedure, to promote animal welfare and minimize stress. After the experimental procedures, the animals were returned to their box for 24 h until the MRI scan and final brain proteomic analysis.

Surgical procedures and MRI were performed under sevoflurane anesthesia (6% induction and 4% maintenance, in a mixture of 70% nitrous oxide and 30% oxygen). Rectal temperature was maintained at 37 ± 0.5 °C using a feedback-controlled heating pad (Neos Biotec, Pamplona, Spain). The preoperative glucose levels were similar in all animals, ranging from 180 to 220 mg/dL. At the end of the procedure, the rats were sacrificed under deep anesthesia (8% sevoflurane).

Analgesia was administered by the subcutaneous injection of buprenorphine (0.01–0.05 mg/kg; Buprelab, 0.3 mg/mL, Labiana Pets, Madrid, Spain) in all the study groups before the surgical procedures.

### 4.2. Study Groups and Surgical Procedures

Eighteen animals were used in this study (*n* = 6 per group). The animals were randomized into the control, sham, and ischemic groups.

Control group: Animals exposed to sevoflurane for 150 min (time corresponding to that used to perform surgical procedure, occlusion time employed to induce ischemia, and MRI analysis).

Sham group: Animals subjected to the same surgical procedure as the ischemic animals, without intraluminal occlusion of the MCA.

Ischemic group: Animals with transient focal ischemia induced by the intraluminal middle cerebral artery occlusion (tMCAO) model (an MCAO surgical video is included as a Appendix A). Transient focal ischemia was induced by intraluminal MCA occlusion as previously described [35], using commercially available sutures with silicone rubber-coated heads (350 μm in diameter and 1.5 mm long; Doccol, Sharon, MA, USA). A medial incision was made in the ventral surface of the neck under a surgical microscope, and the digastric muscles were dissected to expose the carotid arterial system, i.e., the CCA, the ECA, and the ICA. The contralateral CCA was exposed, and a 6/0 suture (Ethicon Mersilk Sutures, Raritan, NJ, USA) was passed without ligation. The CCA on the ipsilateral side was ligated with a 6/0 suture immediately before the bifurcation of the ECA and ICA. The ECA and ICA were then exposed and ligated using 6/0 sutures. The ECA branches were coagulated. The pterygopalatine artery (PPA) was identified and ligated with a 6/0 suture near its origin to prevent the filament from entering the branch. A small incision was made in the vascular segment between the CCA-ECA bifurcation and the most distal end, where the ACE suture was performed to avoid cutting the artery completely. Through this incision, the filament was introduced and advanced through the ACE and ICA until the bifurcation of the pterygopalatine artery (PPA). Cerebral blood flow was monitored using a Periflux 5000 laser Doppler perfusion monitor (Perimed AB, Järfälla, Sweden) by placing a Doppler probe (model 411; Perimed AB, Järfälla, Sweden) under the temporal muscle at the parietal bone surface near the sagittal crest. The contralateral CCA was then ligated, and the filament was carefully passed through the ICA until it reached the MCA, as indicated by Doppler signal reduction (Appendix A). Once artery occlusion was achieved, each animal was carefully moved from the surgical bench to the magnetic resonance (MR) system for MR angiography (MRA), to ensure that the artery remained occluded throughout the MR procedure and to detect possible arterial malformations [36,37]. After a basal MR analysis, the animals were returned to the surgical bench, and the Doppler probe was repositioned. Reperfusion was performed 75 min after the onset of occlusion. In line with our previous studies using the same ischemic model, the following exclusion criteria were used [38]: (1) <70% reduction in relative cerebral blood flow during arterial occlusion; (2) arterial malformations, as determined by MRA; and (3) the absence of reperfusion or prolonged reperfusion (>10 min until achieving ≥50% of baseline cerebral blood flow) after filament removal. MRI-T2 scans for infarct assessment were determined 1 day after ischemia.

Cerebral blood flow was also monitored in control and sham animals using laser Doppler, following the same protocol as described for ischemic animals. The protocol used in this study is summarized in Appendix A.

The sample size/group was calculated based on the variance obtained in our previous study, which was developed to optimize the inclusion criteria for the ischemic model used in this study [38].

### 4.3. MRI and Image Analysis

MRI studies were conducted on a 9.4 T horizontal bore magnet (Bruker BioSpin, Ettlingen, Germany) with 12 cm wide actively shielded gradient coils (440 mT/m). Radiofrequency transmission was achieved using a birdcage volume resonator, and the signal was detected using a four-element arrayed surface coil positioned over the head of the animal. The latter was fixed using a tooth bar, earplugs, and adhesive tape. The transmission and reception coils were actively decoupled from each other. Gradient echo pilot scans were performed at the beginning of each imaging session to position the animals accurately inside the magnetic bore. MRI post-processing was performed using the ImageJ software (Version 1.8.0). Infarct volumes were determined from T2 relaxation maps by the manual selection of areas of hyperintense T2 signals by a researcher blinded to the animal protocols. The infarct size was defined as the percentage of ischemic damage with respect to the ipsilateral hemispheric volume corrected for brain edema. For each brain slice, the total areas of both hemispheres and the areas of infarction were calculated. The edema index was measured by quantifying the midline deviation (MD), which was calculated as the ratio between the volumes of the ipsilateral and contralateral hemispheres. The actual infarct size was adjusted for edema by dividing the infarction area by the edema index [mm^3^/MD]. The infarct volume was calculated as follows: (infarct volume [mm^3^/MD]/ipsilateral hemispheric area [mm^3^]) × 100. These procedures have been repeatedly used to measure and evaluate stroke outcomes in experimental models [39,40,41].

MR angiography: A non-invasive angiography was performed using time-of-flight magnetic resonance angiography (TOF-MRA), as reported previously [36,38]. TOF-MRA scans were performed with a 3D-Flash sequence with an ET = 2.5 ms, RT = 15 ms, FA = 20°, NA = 2, SW = 98 KHz, 1 slice of 14 mm, 30.72 × 30.72 × 14 mm^3^ field of view (FOV) (with saturation bands to suppress signals outside this FOV), and a matrix size of 256 × 256 × 58 (resolution of 120 µm/pixel × 120 µm/pixel × 241 µm/pixel), and implemented without the fat suppression option.

MRI T2 maps: Ischemic lesions were determined from T2 maps calculated from T2-weighted images acquired 24 h after the experimental intervention in the three experimental groups (control, sham, and ischemia). The following MSME sequence was used: with an ET = 9 ms, RT = 3 s, 16 echoes with 9 ms echo spacing, a flip angle (FA) = 180°, NA = 2, spectral bandwidth (SW) = 75 KHz, 14 slices of 1 mm, 19.2 × 19.2 mm^2^ FOV (with saturation bands to suppress signals outside this FOV), and a matrix size of 192 × 192 (isotropic in-plane resolution of 100 µm/pixel × 100 µm/pixel), and implemented without the fat suppression option.

The MRI protocol used in the ischemic group was also performed on control and sham animals. The MR and MRI T2 mapping durations were 20 and 40 min, respectively.

### 4.4. Qualitative and Quantitative Proteomic Analysis in Brain Tissue

#### 4.4.1. Perfusion and Tissue Processing

The animals were deeply anesthetized with sevoflurane (6% in a mixture of 70% NO_2_ and 30% O_2_) and transcardially perfused with 100 mL of 0.1 M PBS (pH 7.4) 24 h after the surgical procedure. The brains were carefully removed from the skull and sectioned at 2 mm thickness using a matrix. Tissue sampling was performed in the 0.2 mm posterior caudal slice in both hemispheres, and samples were collected from the peri-infarct and infarct areas (Appendix A) of both hemispheres. Data acquisition was performed individually for each sample, and data analysis was conducted by pooling samples from each brain. The tissue was stored at −80 °C for further analysis.

#### 4.4.2. Protein Extraction and Digestion

Frozen tissue (100 mg) from the different brain samples was homogenized in 300 μL RIPA buffer [200 mmol/L Tris/HCl (pH 7.4), 130 mmol/L NaCl, 10% (*v*/*v*) glycerol, 0.1% (*v*/*v*) SDS, 1% (*v*/*v*) Triton X-100, and 10 mmol/L MgCl_2_] with antiproteases and antiphosphatases (Sigma-Aldrich, St. Louis, MO, USA) in a TissueLyser II (Qiagen, Tokyo, Japan). The homogenate was centrifuged at 14,000× *g* at 4 °C for 20 min. The protein concentration was measured using an RC-DC kit (Bio-Rad Laboratories, Hercules, CA, USA) according to the manufacturer’s protocol. Protein aliquots of 100 μg were concentrated in a single SDS-PAGE band [11,42] and submitted to a manual digestion as previously described [42]. Finally, after peptide extraction using 50% (*v*/*v*) ACN/0.1% (*v*/*v*) TFA (×3) and ACN (×1), the peptides were pooled, concentrated in a SpeedVac system (Thermo Fisher Scientific, Waltham, MA, USA), and stored at −20 °C.

#### 4.4.3. Qualitative (LC-MS/MS) Analysis

The digested peptides from each sample were separated using reverse-phase chromatography for protein identification. The gradient was developed using a micro-LC system (Eksigent Technologies nanoLC 400, Sciex, Redwood City, CA, USA) coupled to a high-speed Triple TOF 6600 mass spectrometer (Sciex, Redwood City, CA, USA) with a microflow source. The analytical column used was a Chrom XP C18 silica-based reversed-phase column (150 0.30 mm) with a 3 mm particle size and 120 Å pore size (Eksigent, Sciex, Redwood City, CA, USA). The trap column was a YMCTRIART C18 (YMC Technologies Teknokroma Analítica, Barcelona, Spain), with a 3 mm particle size and 120 Å pore size, which was switched on-line with the analytical column. Data were acquired using a TripleTOF 6600 system (Sciex, Redwood City, CA, USA) with a data-dependent workflow (DDA). The micro-pump generated a flow rate of 5 µL/min and was operated under gradient elution conditions, using 0.1% formic acid in water as mobile phase A and 0.1% formic acid in acetonitrile as mobile phase B. Peptides were separated using a 90 min gradient ranging from 2% to 90% mobile phase B [43,44].

Data were acquired using a TripleTOF 6600 System (Sciex, Redwood City, CA, USA) with a DDA workflow. The source and interface conditions were as follows: ion spray floating voltage (ISVF), 5500 V; curtain gas (CUR) 25, collision energy (CE), 10; and ion source gas 1 (GS1), 25. The instrument was operated using Analyst TF 1.7.1 software (Sciex, Redwood City, CA, USA). The switching criteria were set to ions greater than a mass-to-charge ratio (*m*/*z*) of 350 and smaller than *m*/*z* 1400, with a charge state of 2–5, a mass tolerance of 250 ppm, and an abundance threshold of more than 200 counts per second (cps). The target precursor ions were excluded after 15 s. The instrument was automatically calibrated every 4 h using tryptic peptides from PepCalMix (Sciex, Redwood City, CA, USA) as an external calibrant [45].

#### 4.4.4. Data Analysis

After the MS/MS analysis (MS2 data), the data files were processed using ProteinPilot 5.0.1 software from Sciex, which uses the algorithm Paragon for database searches and Progroup for data grouping. Data were searched for using the rattus-specific UniProt database, specifying iodoacetamide at cysteine alkylation as a variable modification and methionine oxidation as a fixed modification. The FDR was determined using a nonlinear fitting method, displaying only the results that reported a 1% global FDR or better [46]. The dysregulated proteins were selected using a *p* < 0.05 and FC > 1.5 or <0.6 as the cut-off in the SWATH analysis.

Scaffold (version Scaffold-5.2.2, Proteome Software Inc., Portland, OR, USA) and Scaffold DDA (version Scaffold DDA-6.4.1, Proteome Software Inc., Portland, OR, USA) were used to perform the DDA analysis, as described previously [43].

#### 4.4.5. Generation of Reference Spectral Library

A pool from each group was analyzed using the shotgun DDA approach. The samples were separated in the micro-LC system Ekspert nLC425 (Eksigen, Dublin, CA, USA), using a Chrom XP C18 150 mm × 0.30 mm, 3 mm particle size, and 120 Å pore size (Eksigen, Dublin, CA, USA) at a flow rate of 10 μL/min, using as a solvent A water, 0.1% formic acid (FA), and solvent B acetonitrile (ACN), 0.1% FA. The peptide separation gradient was from 5% to 95% B for 30 min, 90% B for 5 min, and 5% B for 5 min for column equilibration, for a total time of 40 min. A 6600 + hybrid quadrupole-TOF mass spectrometer (Sciex, Redwood City, CA, USA) was coupled with the LC. A 250 ms survey scan was performed from 400 to 1250 *m*/*z* using a mass spectrometer, followed by MS/MS experiments from 100 to 1500 *m*/*z* (acquisition time of 25 ms) for a total cycle time of 2.8 s. The fragmented precursors were added to the dynamic exclusion list for 15 s, and any ion with a charge +1 was excluded from the MS/MS analysis. Protein identification was performed using ProteinPilot software v.5.0.1. (Sciex, Redwood City, CA, USA) using the Rattus norvegicus-specific UniProt Swiss-Prot database. The FDR was set to 1 for peptides and proteins, with a confidence score above 99% [46].

#### 4.4.6. Quantification by SWAT-MSH and Data Analysis

A quantitative proteomic analysis was performed using the SWATH method on a hybrid quadrupole-TOF mass spectrometer, 6600+ (Sciex, Redwood City, CA, USA), as previously described [47,48,49]. SWATH–MS acquisition was performed using an IDA (independent data analysis) method. Protein from each sample (4 µg) was subjected to chromatographic separation, as described previously. The SWATH–MS method is based on repeating a cycle consisting of the acquisition of 100 TOF MS/MS scans (400–1500 *m*/*z*, high-sensitivity mode, 50 ms acquisition time) of overlapping sequential precursor isolation windows of variable width (1 *m*/*z* overlap) covering the 400–1250 *m*/*z* mass range, with a previous TOF MS scan (400–1500 *m*/*z*, 50 ms acquisition time) for each cycle. The total cycle time was 6.3 s. For each sample set, the width of the 100 variable windows was optimized according to the ion density found in the DDA runs using the SWATH–MS variable window calculator worksheet from Sciex.

The targeted data extraction of the fragment ion chromatogram traces from the SWATH–MS runs was performed by PeakView (version 2.2) using the SWATH–MS Acquisition MicroApp (version 2.0). This application processed the data using the spectral library created from the DDA data loaded over this library of individual samples acquired using the SWATH method. Up to ten peptides per protein and seven fragments per peptide were selected based on the signal intensity to obtain the peak areas, and any shared and modified peptides were excluded from processing.

Integrated peak areas (SWATH–MS areas) were directly exported to MarkerView software (Version 1.3) (Sciex, Redwood City, CA, USA) for a relative quantitative analysis. MarkerView uses processing algorithms that accurately identify chromatographic and spectral peaks directly from raw SWATH data. First, the integrated peak areas were normalized using MLR normalization or suma total areas based on the analysis performed, and an unsupervised multivariate statistical analysis using principal component analysis (PCA) was performed, to compare the data across the samples using scaling. Student’s *t*-test analysis was performed using MarkerView software to compare the samples. The dysregulated proteins were selected using *p* < 0.05 and FC > 1.5 or <0.6 as a cut-off. Individual values of the SWATH–MS areas per protein and sample were used to create box pots.

#### 4.4.7. Protein Functional Enrichment and Network Analysis

Differentially regulated proteins were subjected to functional analysis and interpreted through various open-access bioinformatics tools for the analyzing of biological information related to molecular functions, biological processes, cellular components, protein classes, pathways, and networks among large and complex datasets. For the functional enrichment and interaction network analysis, we used STRING (Version: 12.0) and FunRich (Version 3.1.4) software. FunRich uses hypergeometric tests and the Benjamini–Hochberg and Bonferroni methods [50].

## Figures and Tables

**Figure 1 ijms-25-07538-f001:**
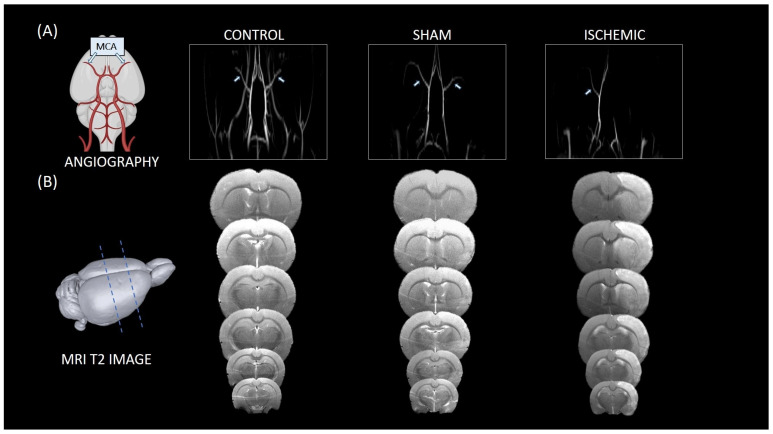
Magnetic resonance imaging (MRI) analysis of the brain. (**A**) Cerebrovascular anatomy of rat. Coronal projection of magnetic resonance angiography (MRA) image of control, sham, and ischemic brains of rats. (**B**) MRI T2 scans of control, sham, and ischemic brains of rats at 24 h after surgical procedure or middle cerebral artery occlusion (MCAO). The figure was created using BioRender.com. Six animals were included in each comparison group.

**Figure 2 ijms-25-07538-f002:**
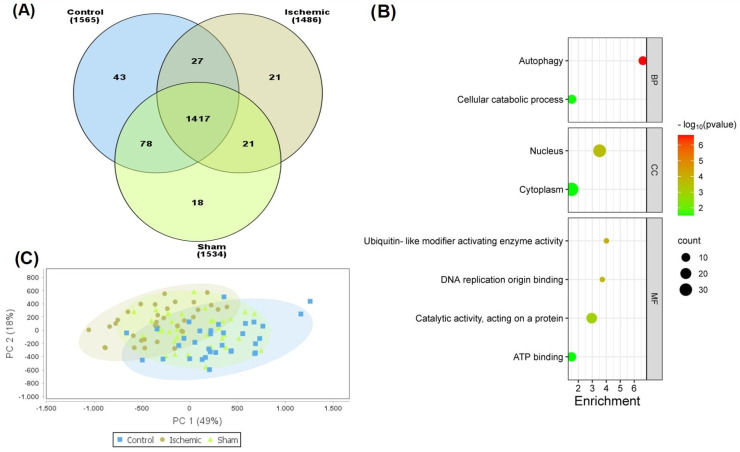
(**A**) A Venn diagram showing the distribution of the identified proteins in the three study groups. (**B**) A schematization of the functional protein association network (BP, biological processes; CC, cellular components; and MF, molecular function) in which the 21 common proteins between the sham and ischemic group are involved. (**C**) Principal components analysis (PCA). Principal component 1 (PC1) and principal component 2 (PC2) were identified by variance in the data-dependent acquisition (DDA). The percentage of variance indicates how much variance was explained by PC1 and PC2. (**A**,**C**) were performed using Scaffold DDA. (**B**) was created using SRplot (a free online platform for data visualization and graphing). Six animals were included in each comparison group.

**Figure 3 ijms-25-07538-f003:**
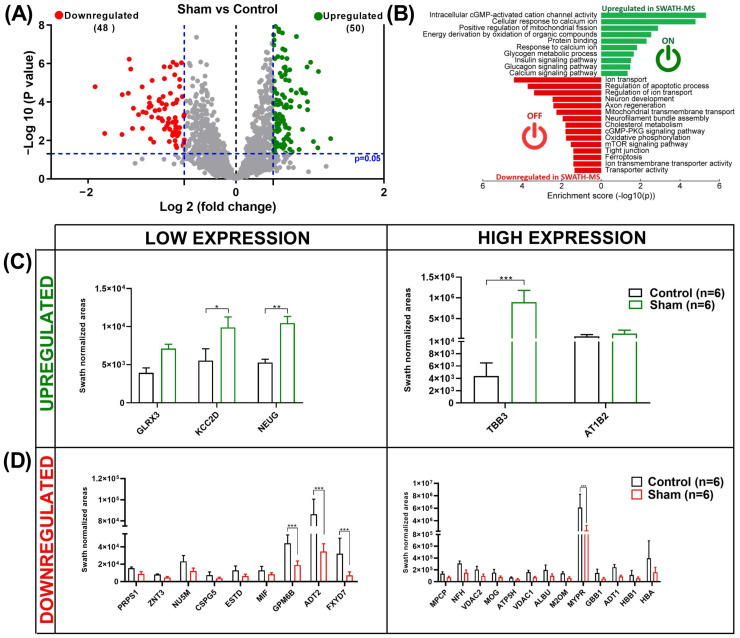
Dysregulated proteins in sham vs. control group. (**A**) Volcano plot resulting from comparison of sham and control groups. Cut-off for protein expression changes was *p* < 0.05, and fold change (FC) > 1.5 or <0.6. Up- and downregulated dysregulated proteins are represented as green and red dots, respectively. Proteins that were not differentially expressed are presented as gray dots. (**B**) STRING interaction analysis of dysregulated proteins: Up- and downregulated dysregulated proteins are represented as green and red bars, respectively. (**C**) The 5 upregulated and (**D**) 22 downregulated proteins that exhibited the greatest fold change in expression in the sham vs. control group. The data are presented as the mean ± standard deviation (SD) of the mean. Statistical analysis was performed using two-way variance (ANOVA) test, followed by Sidak’s multiple comparison test: * (*p* < 0.05), ** (*p* < 0.01), *** (*p* < 0.001). (**B**) was created using SRplot (a free online platform for data visualization and graphing). Six animals were included in each comparison group.

**Figure 4 ijms-25-07538-f004:**
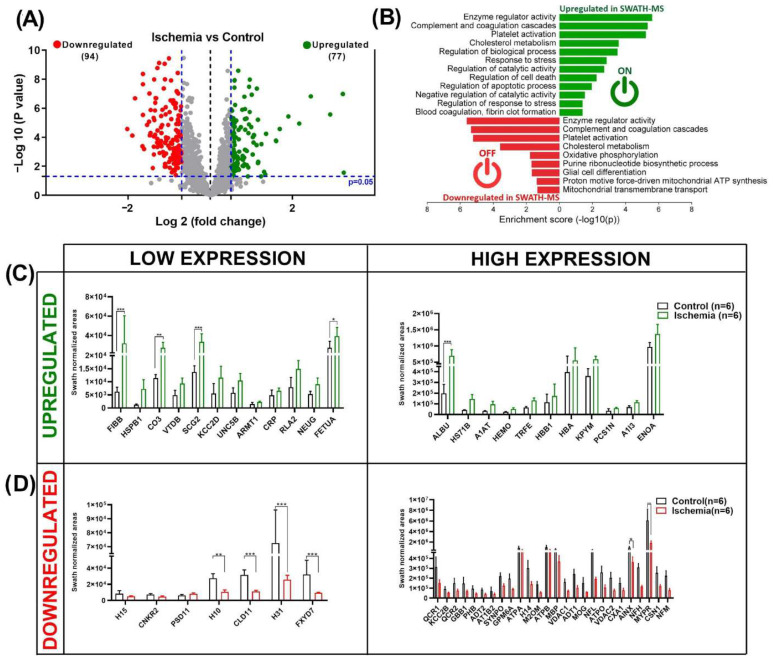
Dysregulated proteins in ischemia vs. control group. (**A**) Volcano plot resulting from comparison of ischemia and control groups. Cut-off for protein expression changes was *p* < 0.05, and FC > 1.5 or < 0.6. Up- and downregulated dysregulated proteins are represented as green and red dots, respectively. Proteins that were not differentially expressed are presented as gray dots. (**B**) STRING interaction analysis of dysregulated proteins: Up- and downregulated dysregulated proteins are represented as green and red bars, respectively. (**C**) The 24 upregulated and (**D**) 33 downregulated proteins that exhibited the greatest FC in expression in the ischemia vs. control group. The data are presented as the mean ± standard deviation (SD) of the mean. Statistical analysis was performed using two-way variance (ANOVA) test, followed by Sidak’s multiple comparison test: * (*p* < 0.05), ** (*p* < 0.01), *** (*p* < 0.001). (**B**) was created using SRplot (a free online platform for data visualization and graphing). Six animals were included in each comparison group.

**Figure 5 ijms-25-07538-f005:**
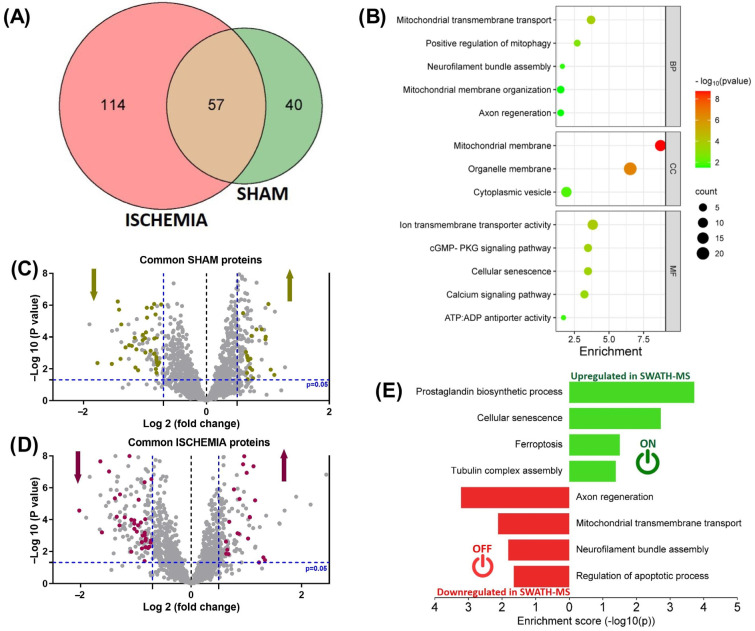
Common dysregulated proteins between the sham and ischemic groups. (**A**) A Venn diagram showing the overlap of common dysregulated proteins identified by SWATH between the sham and ischemic groups. (**B**) A schematization of the functional protein association network (BP, biological processes; CC, cellular components; and MF, molecular function) in which the fifty-seven common proteins identified by SWATH between the sham and ischemic groups were involved. (**C**,**D**) Results of dysregulated proteins presented corresponding to sham vs. control (**C**) and ischemia vs. control (**D**) using volcano plots. Cut-off point for protein expression changes was *p* < 0.05, and FC > 1.5 or <0.6. Common dysregulated proteins in the sham group are represented as brown dots and in the ischemic group as purple dots. (**E**) STRING interaction analysis of dysregulated proteins: Up- and downregulated dysregulated proteins are represented as green and red bars, respectively. (**A**) was created using the FunRich program. (**B**,**E**) were created using SRplot (a free online platform for data visualization and graphing). Six animals were included in each comparison group.

## Data Availability

Mass spectrometry proteomics data have been deposited in the ProteomeXchange Consortium via the PRIDE [51] partner repository with the dataset identifier PXD052899.

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
