# Peer review of "Comparative Brain Proteomic Analysis between Sham and Cerebral Ischemia Experimental Groups"

_ijms, 2024, doi:10.3390/ijms25147538_

Round 1
Reviewer 1 Report (New Reviewer)
Comments and Suggestions for Authors
The manuscript entitiled Comparative brain proteomic analysis between sham and cerebral ischemia experimental groups describes very important issue i.e., the use of the sham group in experimental studies of brain ischemia. As the Authors demonstrated, there are many significant changes in protein levels in sham animals compared to control animals, which should be taken in consideration during results interpretation. The introduction provides sufficient informations, the methods, results and discussion are described well.
The only question that I have is:
Did the Authors try to confirm the most significant results obtained through LC-MS/MS with other methods? For example with Western Blot or ELISA?

Author Response
Comments and Suggestions for Authors : The manuscript entitiled Comparative brain proteomic analysis between sham and cerebral ischemia experimental groups describes very important issue i.e., the use of the sham group in experimental studies of brain ischemia. As the Authors demonstrated, there are many significant changes in protein levels in sham animals compared to control animals, which should be taken in consideration during results interpretation. The introduction provides sufficient informations, the methods, results and discussion are described well.
The only question that I have is: Did the Authors try to confirm the most significant results obtained through LC-MS/MS with other methods? For example with Western Blot or ELISA?
Response to Reviewer #1: We appreciate the Referee's positive comments and are pleased that the revised manuscript has clarified the initial limitations of the study.
Regarding the question whether the most significant results by LC-MS/MS were obtained with other methods, such as Western blot or ELISA. This is an interesting point to address in future studies. As we stated in the discussion, we did not perform this validation because, the main aim of the study was to evaluate how surgical intervention of the MCAO model affects the protein profile that could lead to an overestimation of the molecular mechanisms caused by the ischaemic damage.
Our future directions in this line are to select those proteins with higher significant fold-changes that appear exclusively increased in the ischemic region (none in the sham group), and validate the expression by WB and immunofluorescence microscopy. In addition, we would like to evaluate the potential utility of these selected proteins as therapeutic targets or suitable biomarkers for targeting the peri-infarct region, with an important value for theranostics (see our previous study in Theranostics. 2013 Dec 12;4(1):90-105).
English language has been checked by an external English language editing service (Editage).
Reviewer 2 Report (New Reviewer)
Comments and Suggestions for Authors
This manuscript reports on the implementation of control (sham) groups in the study of brain ischemia. The intraluminal filament procedure, a common model of middle cerebral artery occlusion (MCAO), was examined with respect to proteomic profiling of sham and the MCAO ischemic group. It was found that approx. 57 proteins that appeared dysregulated in both sham and ischemic animals, but also proteins were dysregulated in the sham group only.
The manuscript provided here is a revision of a previous submission and it now appears well-described and generally reads very well.
Author Response
Reviwer Comments: This manuscript reports on the implementation of control (sham) groups in the study of brain ischemia. The intraluminal filament procedure, a common model of middle cerebral artery occlusion (MCAO), was examined with respect to proteomic profiling of sham and the MCAO ischemic group. It was found that approx. 57 proteins that appeared dysregulated in both sham and ischemic animals, but also proteins were dysregulated in the sham group only.The manuscript provided here is a revision of a previous submission and it now appears well-described and generally reads very well. Response to Reviewer #2: We are pleased that the revised manuscript meets the requirements for publication.Reviewer 3 Report (New Reviewer)
Comments and Suggestions for Authors
The article titled Comparative brain proteomic analysis between sham and cerebral ischemia experimental Groups conducted by Lourido et al. provides important information to improve the validation and quality of animal experimental models.
before publishing, you must take into consideration the following points
- Line 34 should be improved
- Lines 162-163 should be written as “cellular component”
- Line 281 please be assured of this sentence proteins was statistically significantly decreased compared to the control
- Line 682 and 683 (control, sham and control), double check this sentence
- line 702-703 the sentence must be improved (the animals time)
- Line 752 (before to) must be checked
- Insert the references, LINE 647 TO 766
- Line 853 is it 80 or -80 °C
- Insert the references to section 4.4.3 Qualitative (LC-MS/MS) Analysis
- Section 4.4.6 Quantification by SWAT-MSH and data analysis, the significance of dysregulated protein (p-value) should also be included in Data analysis section 4.4.4
- Do you think that the proteins affected by the surgical procedure have a direct effect on the ischemia pathway?
Comments on the Quality of English LanguageMinor editing of English language required
Author Response
Reviwer Comments: The article titled Comparative brain proteomic analysis between sham and cerebral ischemia experimental Groups conducted by Lourido et al. provides important information to improve the validation and quality of animal experimental models. Before publishing, you must take into consideration the following points:
- Line 34 should be improved
- Lines 162-163 should be written as “cellular component”
- Line 281 please be assured of this sentence proteins was statistically significantly decreased compared to the control
- Line 682 and 683 (control, sham and control), double check this sentence
- line 702-703 the sentence must be improved (the animals time)
- Line 752 (before to) must be checked
- Insert the references, LINE 647 TO 766
- Line 853 is it 80 or -80 °C
- Insert the references to section 4.4.3 Qualitative (LC-MS/MS) Analysis
- Section 4.4.6 Quantification by SWAT-MSH and data analysis, the significance of dysregulated protein (p-value) should also be included in Data analysis section 4.4.4
- Do you think that the proteins affected by the surgical procedure have a direct effect on the ischemia pathway?
Response to Reviewer #3: We appreciate the Referee's positive words and are pleased that the revised manuscript has clarified the initial limitations of the study.
In line with the Reviewer #1 comments, whether the proteins affected by the surgical procedure have a direct effect on the ischemia pathway. In the discussion, we have already commented that, based on previous studied, some of the detected proteins were involved in ischemia pathway (See supplemental table 2 and Figure 3A). Our future directions in this line are to select those proteins with higher significant fold-changes that appear exclusively increased in the ischemic region (none in the sham group), and validate the expression by WB and immunofluorescence microscopy. We would like also to evaluate the potential utility of these selected proteins as therapeutic targets or suitable biomarkers for targeting the peri-infarct region, with an important value for theranostics in the in cerebral ischaemia (see our previous study in Theranostics. 2013 Dec 12;4(1):90-105).
English language has been checked by an external English language editing service (Editage).
This manuscript is a resubmission of an earlier submission. The following is a list of the peer review reports and author responses from that submission.
Round 1
Reviewer 1 Report
Comments and Suggestions for Authors
The authors aim to highlight the changes on brain protein expression that are not associated with the ischemic lesion in common models of middle cerebral artery occlusion (MCAO). Although this is a relevant issue, this study looks rather preliminary to support the main conclusions reached for the reasons indicated below. On these grounds, I cannot recommend publication of this article in its present form.
Major concerns:
1. The number of experimental animals used per experimental condition (3) is too short for a reliable statistical analysis. It is recommended to increase this number to 6-8 animals per experimental condition.
2. The changes of protein expression induced solely by the 120 min treatment with sevoflurane (anesthesia treatment) should also be included and commented, for a proper evaluation of stress-linked protein expression changes during experimental handlings of these animals. This is particularly relevant in this work, since the number of proteins displaying altered expression levels is very large and in many cases with modest fold-change (FC) values.
3. It is shocking that the changes of hemoglobin isoforms HBA, HBB1 and HBB2, which are among those showing the large changes of expression (if not the largest change), are larger in sham vs. control animals than in ischemic versus control animals. Also, it is puzzling that only “with the exception of SYUG and TCTP proteins, all other proteins are dysregulated in the same way in the sham and ischemic groups” (page 5, lines 147-148). Of note, the reported change of TCTP expression in ischemic versus control group is very modest, log 2(FC) = 0.1702.
4. The standard deviation of the log 2(FC) values of each one of the dysregulated proteins are not given (n = 6-8 animals) and should be included in the tables. These data are needed for a critical evaluation of the acceptable FC cut-offs values in this work.
Reviewer 2 Report
Comments and Suggestions for Authors
The article has a potential interest, and it is important for animal research. Nevertheless, several vital points must be addressed before final acceptance in the International Journal of Molecular Sciences.
Method part:
· How about control? Were you subjected to the same MRI imaging protocol described for the ischemic group? The picture has, but the procedure description does not.
· How many animals lived in one box, how long would you keep them together?
· Did you give any medications for pain reduction? It is common practice to give additional pain relievers. If given, did you inject the control group as well?
· How did you choose the number of rats in the group? Did you calculate the power analysis?
· It is unclear when the brain tissue for proteomic analysis was collected; please clarify! In the method and material part “4.3. Magnetic resonance imaging and image analysis”, it was written: “MRI T2-maps: ischemic lesions were determined from T2-maps calculated from T2 weighted images acquired 24 hours, 7 and 14 days after the onset of ischemia using a MSME sequence…..” At the same time, in the results part, you wrote: “Twenty-four hours after surgery, the brain tissue of the three experimental groups (control, sham and ischemic group) were analyzed by proteomic analysis”.
· Can you clarify which part of the brain, section, was sampled for the proteomics analyses? It is known that the amount and activity of different receptors differ in different brain hemispheres, which may affect the result. Were both hemispheres - ipsilateral and contralateral – used together for proteomic analysis?
Results
· Can you include the results of infarct volume?
· Image B of Figure 2 should be improved by increasing the font size of the text.
· Does picture C of Figure 4 have the same group colour transcripts as picture D?
· The font sizes for the figures should be improved and increased!
· Could you name the 17 altered proteins for all three groups (Figure 5D)?
· Were any behavioural changes observed in the rats after the stroke? Did you evaluate that?
· It will be beneficial to compare and add results of ipsilateral and contralateral sides from ischemic rats.
· How about blood samples? Can you do a proteomic analysis of blood samples from all three groups and compare them with brain samples? It will be crucial because brain tissue samples from the patients are complex to get, but animal studies can help understand which pathways and markers could be helpful in therapy.
· Could you show the variability of the samples within the same group and plot the graphs with individual values as well?
Discussion
· It is necessary to supplement the discussion with a section on which proteins should be determined and their activity measured to judge the effectiveness of the therapy.
· In addition, discuss whether the significantly changed proteins in your study also match the measurements made in the clinic for patients. Does the altered protein profile in animal brain tissue correlate with the protein profile in the blood plasma of patients after stroke?
Is current therapy targeting the significantly altered protein profile?

Reviewer 3 Report
Comments and Suggestions for Authors
In their investigation, the authors have addressed a pivotal aspect of experimental stroke research: incorporating a sham-operated control group. Given the methodological and foundational nature of this study, detailed documentation of the MCAO (middle cerebral artery occlusion) procedure is imperative. Merely citing previous protocols does not suffice; a comprehensive description of the entire surgical sequence is warranted, with a supplemental video protocol strongly advised. Detailed attention to operative specifics, including the potential cauterization of collateral arteries such as the Pterygopalatine artery, is required.
In the 'Materials and Methods' section, it is necessary to note whether the intact group underwent an MRI procedure with anesthesia, similar to the ischemia group, since additional anesthesia could also have affected the protein expression profile.
Specify under what anesthesia the MRI investigation was conducted and its duration.
Further, it is imperative to clarify whether the sham group was subjected to Doppler perfusion monitoring as well.
In the 'Materials and methods' section, you have to indicate from which area of the brain the tissue was taken for the study and at what time after modeling. For a convincing demonstration of the influence of sham surgery, brain tissue should be taken from both hemispheres and from brain regions supplied by different vessels (e.g. the anterior and middle cerebral artery).
It is recognized that protein expression profiling is an exceedingly sensitive technique; thus, the expertise of the experimenter in performing the MCAO model could be a variable of consequence. It is of paramount importance to ensure that the observed trends are not artifacts of experimental handling and, therefore, replication of the experiments by an independent experimenter is advised for validation.
Round 2
Reviewer 1 Report
Comments and Suggestions for Authors
This manuscript addresses a relevant conceptual concept for the correct interpretation of the results obtained in MCAO animal models. Unfortunately, the number of samples used per experimental group is too low (as pointed out in my report to the previous version of this manuscript), and this has not been improved. Therefore, the contributions of interindividual variability to the reported protein expression changes have been insufficiently evaluated. Due to this, the statistical significance of many of the claimed protein expression changes and conclusions of this work are highly questionable, as noted in my previous report to the first version of this manuscript. In my opinion, the data shown in this work do not allow to reach solid scientific conclusions, as commented in more detail below. On these grounds, I cannot recommend publication of this manuscript.
Major points:
-Number of animals per experimental condition: The arguments of the authors’ response letter are puzzling, as the proteomic analysis using the same SWATH-MS method with a larger number of samples per experimental condition, in most cases n ≥6, have been performed in the more recent studies cited in the author’s response (Mol Cell Proteomics. 2022 Dec;21(12):100435; Cell Metab. 2023 Sep 5;35(9):1630-1645.e5; Mol Ther Nucleic Acids. 2023 Mar 21; 32:247-262; Brain Behav Immun. 2023 Oct; 113:44-55).
-Standard errors of protein determinations: The standard errors of protein determinations shown in Figures 3C, 3D, 4C and 4D cast serious doubt on the statistical significance of the protein expression changes observed for many of the proteins that are claimed to be dysregulated in the main text of the Results section. Note that the difference of protein expression between sham and control groups is within the range of the standard error bars for PI42C, HNRPL, HPCL4 and TBA4A in the Figure 3C; NFH and S6A11 in the Figure 3D; SYUG, FRIL1, ARP5L, MGLL, PCP4 and NCKP1 in the Figure 4C; and ACADL in the Figure 4D. Also, the large error bar size observed in some cases recommends to accumulate data from more samples to reach a solid conclusion, e.g., NFH (Sham-Figure 3D) and RTN1 (Ischemia-Figure 4C).
-Criteria of cut-off minimum Fold-Change (FC) of protein. The following sentence of the author’s response: “the fold change is in the order of other fold changes obtained in numerously papers” is ambiguous, either intentionally or not. The values set in this manuscript are 1.5 (up-regulation) or 0.8 (down-regulation), while it was 2.0 (up-regulation) or 0.5 (down-regulation) in a recent paper in where several authors of this manuscript were co-authors (Mol Ther Nucleic Acids. 2023 Mar 21; 32:247-262). When the number of replicates is too low, does not seem a better scientific praxis to follow the criterium of higher cut-off fold-change? Of course, this will largely reduce the number of positive results to only 4 proteins (3 up-regulation +1 down-regulation) in the Table 2 (sham vs. control), 2 proteins (up-regulation) in the Table 3 (ischemic vs. control), 2-3 proteins in the Table 4 (up-regulation) and 3 proteins in the Table 5 (2 up-regulation and 1 down-regulation). Also, it must be noted that most of these positive results are hemoglobin isoforms (HBB1, HBB2 and HBA), and that only two additional proteins fulfil the more stringent cut-off criteria, i.e., EC12 and SYUG. Obviously, this undermines the strong conclusive statement of lines 305-307 : “Our results provide clear evidence that the surgery required to induce the MCA occlusion induces protein expression changes in the brain, which supports the needed of including sham groups in the experimental designs of preclinical studies.” Clearly, this conclusion looks rather speculative with the data included in this manuscript.
Author Response
Dear reviewer, we fully agree with your comments regarding the number of animals and the proteomic analyses.
We have asked the Editor for an extension of three months to complete this additional analysis, we will try to submit the revision as soon as possible.
Sincerely
Francisco Campos, PhD
Clinical Neurosciences Research Laboratory
Translational Stroke Laboratory Group (TREAT)
Health Research Institute of Santiago de Compostela (IDIS)
Clinical University Hospital (CHUS), SERGAS
Travesía da Choupana, s/n, Santiago de Compostela
E15706 A Coruña, Spain
Phone: +34 981951097
Fax: +34 981951086
E-mail: francisco.campos.perez@sergas.es
Reviewer 3 Report
Comments and Suggestions for Authors
According to modern bioethical 3R principles, it is essential for researchers to balance the need to obtain reliable results with minimising the number of animals used in their studies. Traditionally, the sham group has served as a surgical quality control to assess neurological impairment in neuroprotective drug research. For example, a carefully performed MCAO procedure results in significant neurological deficits, as opposed to a sham operation which does not. However, the routine use of a sham-operated group may lead to the unnecessary use of animals and thus violate bioethical standards. Therefore, any decision to use additional animals must be carefully justified on the basis of a comprehensive review of the experimental evidence.
The conclusions of this study are clear regarding the need for sham-operated animals in ischaemia studies. However, the authors have not provided convincing and unequivocal results in support of these basic conclusions. There are a number of significant limitations to this study. One notable limitation of the study is the lack of specific details on the areas affected by the protein profile changes namely the infarct zone and peri-infarct region and the lack of demonstrated correlation with functional outcomes. In addition, the scientific value of the study would be greatly enhanced by demonstrating that the observed changes in the protein profile remain consistent regardless of the operator conducting the procedure.
To summarise, the authors appear to have reached premature conclusions about the significance of their findings. Further research is needed, along with a detailed description of the limitations of the research outcomes.
Author Response

(The authors gave the same response as above.)
